# Learning from Less: Bayesian Neural Networks for Optimization Proxy using Limited Labeled Data

**Parikshit Pareek**[*]
Theoretical Division
Los Alamos National Laboratory
Los Alamos, NM 87545
pareek@lanl.gov

**Kaarthik Sundar**
Information Systems & Modeling Group
Los Alamos National Laboratory
NM, USA 87545
kaarthik@lanl.gov

**Deepjyoti Deka**
MIT Energy Initiative
Massachusetts Institute of Technology
MA, USA - 02139
deepj87@mit.edu

**Sidhant Misra**
Theoretical Division
Los Alamos National Laboratory
Los Alamos, NM 87545
sidhant@lanl.gov

## Abstract

This work introduces a learning scheme using Bayesian Neural Networks (BNNs) to solve constrained optimization problems in a setting with limited labeled data and restricted model training time. We propose a Semi-Supervised BNN for this practical but complex regime wherein training commences in a sandwiched fashion, alternating between a supervised (using labeled data) learning step for minimizing cost, and an unsupervised (using unlabeled data) learning step for enforcing constraint feasibility. Both supervised and unsupervised steps use Bayesian approach where variational inference is used for approximate Bayesian inference. We show that the proposed Semi-supervised learning method outperforms conventional BNN and deep neural network (DNN) architectures for important non-convex constrained optimization problems from energy network operations, with 50% reduction in mean square error (MSE) along with halving of optimality and feasibility gaps without requiring correction or projection steps.

## 1 Introduction

Bayesian Neural Networks (BNNs) attempt to bring the advantages of Bayesian statistics into the function-approximating capabilities of deep neural networks (DNNs) and have found application in areas ranging from medical image segmentation to fluid dynamics [2, 7, 12, 4, 5]. Improvements in underlying algorithms for training and inference have led to better understanding of BNNs [8, 1, 13] and enabled their use as surrogates for Bayesian optimization [11]. In recent years, DNNs have been applied to solve various optimization problems with physics-based constraints on variables, particularly in energy networks [21, 6, 3, 18, 14, 10]. Here, the primary motivation is to replace time-consuming optimization algorithms with ML proxies, enabling instantaneous solutions to problems on large number of instances. While promising in mimicking optimization solvers, they either rely on enormous labeled datasets to train ML models [14] or require time-consuming constraint correction steps within the framework [3, 6, 21]. We propose a novel BNN-based framework to learn optimization proxies with minimal labeled data and within training time constraints. Leveraging

---

[*]Corresponding Author, Current Affiliation: Department of Electrical Engineering, Indian Institute of Technology Roorkee (IIT Roorkee), India. Email: pareek@ee.iitr.ac.in

Submitted to Workshop on Bayesian Decision-making and Uncertainty, 38th Conference on Neural Information Processing Systems (BDU at NeurIPS 2024). Do not distribute.

BNNs' ability to perform with limited data, the semi-supervised approach addresses the challenge of scarce labeled data in optimization problems with uncertainty. Initial results show that our method outperforms standard approaches in low-data regimes, avoids correction steps, and maintains fast prediction speeds, making it suitable for large number of instances.

## 2  Proposed Semi-supervised BNN Learning

Semi-supervised learning methods aim to leverage unlabeled data to improve the performance of ML algorithms under minimal amount of labeled data availability [20]. Approaches in this area include augmenting unlabeled data with cheap pseudo-labels, developing an unsupervised loss function, and minimizing it with the supervised loss function[17, 20]. For example, data augmentation approach has been used before in the context of image classification using the notion of semantic similarity [17]. However, this notion is not readily extensible to ML proxies for constrained optimization problems, where slight variations in input might lead to significant changes in output.

To circumvent aforementioned difficulty, we propose a feasibility-based data-augmentation scheme where feasibility relates to the constraints of the optimization problem. To the best of our knowledge, these ideas have not been explored in the context of BNN algorithms to solve large-scale optimization problems. Though not directly addressing this problem, one related work worth noting is that of loss function-based prior design [16] for output constraint satisfaction [19].

**Problem Setup:** We consider nonlinear constrained optimization problems having both equality $g(\cdot)$ and inequality constraints $h(\cdot)$, with decision $\mathbf{y}$ and input $\mathbf{x}$ variables as vectors.

$$\min_{\mathbf{y}} \quad c(\mathbf{y}) \tag{1}$$

$$\text{s.t.} \quad g(\mathbf{x}, \mathbf{y}) = 0 \tag{2}$$

$$h(\mathbf{x}, \mathbf{y}) \leq 0 \tag{3}$$

Furthermore, we assume that $\forall \mathbf{x} \in \mathcal{X}$, there exists at least one feasible solution for (1). The goal is to develop a BNN surrogate that provides an approximate optimal value of decision variables $\widehat{\mathbf{y}}_t$ for a given test input vector $\mathbf{x}_t \in \mathcal{X}$. Let $\mathcal{D} = \{(\mathbf{x}_i, \mathbf{y}_i^\star)\}_{i=1}^N$ denote the labeled dataset where $\mathbf{y}_i^\star$ is obtained by solving the optimization problem (1) for $\mathbf{x}_i$. We assume inexpensive sampling for input vector $\mathbf{x}$ and construct the unlabeled data set $\mathcal{D}^u = \{\mathbf{x}_j\}_{j=1}^M$.

**BNN Set-up and Training:** Mathematically, we denote the BNN as $f_w(\mathbf{x})$, where $w$ are the weights and biases that follow an isotropic normal prior $p(w)$ with covariance $\sigma^2 I$.
The supervised part of the BNN training aims to compute the posterior distribution over the weights given labeled data $\mathcal{D}$, and is expressed as: $p(w|\mathbf{x}, \mathbf{y}) \propto p(\mathbf{y}|\mathbf{x}, w) \, p(w)$ where $p(\mathbf{y}|\mathbf{x}, w)$ is the likelihood of the labeled data $(\mathbf{x}, \mathbf{y} \in \mathcal{D})$ given the weights, $p(w)$ is the prior over the weights. The posterior distribution $p(w|\mathbf{x}, \mathbf{y})$ encapsulates the uncertainty about the weights after observing the labeled data. Due to the computational challenges of finding the normalization constant, approximate methods such as variational inference (VI) [9] are used to compute the posterior. For predictions, the posterior prediction is approximated as $p(\mathbf{y}^t|\mathbf{x}^t, \mathcal{D}) = \mathbb{E}_{p(w|\mathcal{D})}[p(f_w(\mathbf{x}^t)]$. Moreover, we use Gaussian likelihood $p(\mathbf{y}|\mathbf{x}, w) = \prod_i \mathcal{N}(\mathbf{y}_i|f_w(\mathbf{x}_i), \sigma_s^2)$ with $\sigma_s^2$ being a parameter in VI, controlling the spread of Gaussian around the target values (noise variance) and $\mathbf{x}_i, \mathbf{y}_i \in \mathcal{D}$.

To effectively incorporate the unlabeled data $\mathcal{D}^u$ into the learning process, it is necessary to define a suitable likelihood function. We propose to augment this unlabeled data using the necessary feasibility condition which vector $\mathbf{y}$ must satisfy to be a solution of (1). Consider a function $\mathcal{L}(\mathbf{y}, \mathbf{x})$ which measures the feasibility of a solution candidate $\mathbf{y}$ for a given input $\mathbf{x}$ such that one term measures the equality gap and other term measures one sided inequality gap or violations, with equal emphasis on both, as

$$\mathcal{L}(\mathbf{y}, \mathbf{x}) = \underbrace{\|g(\mathbf{x}, \mathbf{y})\|^2}_{\text{Equality Gap}} + \underbrace{\|\text{ReLU}[h(\mathbf{x}, \mathbf{y})]\|^2}_{\text{Inequality Gap}} \tag{4}$$

For any given feasible solution $\mathbf{y}_c$[2], $\mathcal{L}(\mathbf{y}_c, \mathbf{x}) = 0$ for the given input. Under the consideration that for each input there exist a solution of (1), we can argue that for each input the feasibility gap

---

[2]Not necessarily optimal for (1).

function (4) has optimal value or true label of 0. We can augment the unlabeled dataset $\mathcal{D}^u$ such that it becomes a labeled feasibility dataset i.e. $\mathcal{D}^f = \{\mathbf{x}_j, 0\}_{j=1}^M, 0$ . Now considering that input sampling is cheap, the construction of this labeled feasibility dataset has no additional computational cost. Similar to the supervised data, we can define a Gaussian likelihood for unsupervised training step as $p(\mathcal{L}|\mathbf{x}, w) = \prod_j \mathcal{N}(0|\mathcal{L}(f_w(\mathbf{x}_j), \mathbf{x}_j), \sigma_u^2)$ with noise variance of unsupervised learning $\sigma_u^2$ and $\mathbf{x}_j \in \mathcal{D}^f$.

For obtaining optimization proxy, we parameterize the candidate solution $f_w(\mathbf{x})$, using deep network architectures and use a sandwich style semi-supervised training for the BNN as shown in Figure 1.The idea is to alternatively use labeled dataset $\mathcal{D}$ and augmented feasibility dataset $\mathcal{D}^f$ for cost optimality and constraint feasibility respectively, to update network weights and biases. Further, the Bayesian inference step (*Sup* and *UnSup*) is performed for a fixed number of iterations with total training time being constrained to $T_{max}$. Finally, the prediction of mean estimate $\mathbb{E}_{\mathbf{y}}$ and predictive variance estimate $\mathbb{V}_{\mathbf{y}}$ is done using a unbiased Monte-carlo estimator via sampling 100 weights from the final weight posterior $p_W^m$.

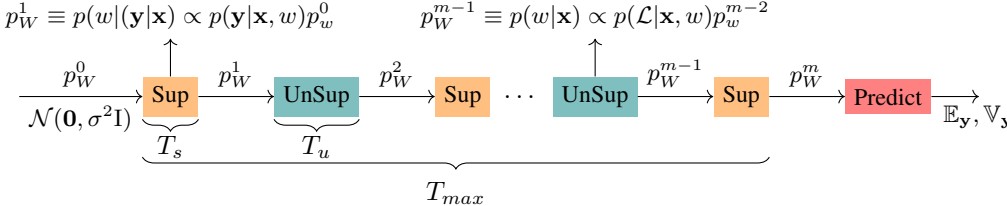

Figure 1: Flowchart of proposed Semi-supervised BNN learning. The *Sup* block represents supervised learning stage with labeled dataset $\mathcal{D}$ and *UnSup* block represents unsupervised learning with augmented feasibility dataset $\mathcal{D}^f$. Learning time upper limits are represented as $T_s$, $T_u$ and $T_{max}$ for *Sup*, *UnSup* and complete Semi-supervised BNN learning respectively.

# 3 Numerical Results: AC Optimal Power Flow

To demonstrate the effectiveness of the proposed semi-supervised learning approach, we focus on the Alternating Current Optimal Power Flow (ACOPF) problem, a crucial decision-making task in electrical power systems. ACOPF aims to determine the least-cost generator set-points while adhering to the operational and physical constraints of the energy network. The problem's inputs are real and reactive power load vectors, and the outputs include generator set-points (real and reactive) and complex node voltages in polar form (magnitude and angle). Variations in the load vector constitute the input dataset $\mathcal{X}$. Furthermore, the mathematical formulation of the ACOPF used in this study represents a non-convex optimization problem, as described in [3]. Additionally, we utilize the publicly available dataset for the 57-Bus system from the DC3 repository [3], for comparative studies. Our neural network architecture has four sub-network of two hidden layers (100 neuron each) with `ReLU` activation function. These four sub-networks are trained to predict real power generation, reactive power generation, voltage magnitude and voltage angle outputs, separately without any overlap. The BNNs are trained using variational inference, utilizing `Numpyro` package while DNNs are trained (with MSE loss over labeled data) using `Pytorch`. All training-testing is performed using a Mac Pro machine with Apple M1 Max processor. We fix $T_s = 30$ sec. and $T_u = 50$ sec. for all Semi-supervised BNN learning instances, following Figure 1. Further, Figure 2 represents the performance of various models with different number of labeled data. All networks have same architecture and best BNN (and DNN) represents the results with hyperparameter optimization (like learning and decay rate). The semi-supervised method uses the best BNN hyperparameters, without any further optimization (details in Appendix A). It is clear that in low labeled data regime, both BNN and proposed Semi-supervised BNN outperforms the DNN approach in terms of MSE errors for various outputs. For feasibility, proposed Semi-supervised method outperforms BNN while DNN's mean equality gap (Eq. Gap) performance improves faster than other methods with increase in number of labeled training samples. This feasibility emphasizing behavior of standard DNN with MSE loss has also been noted in [3], with higher optimality gap as seen in Cost subfigure of Figure 2.

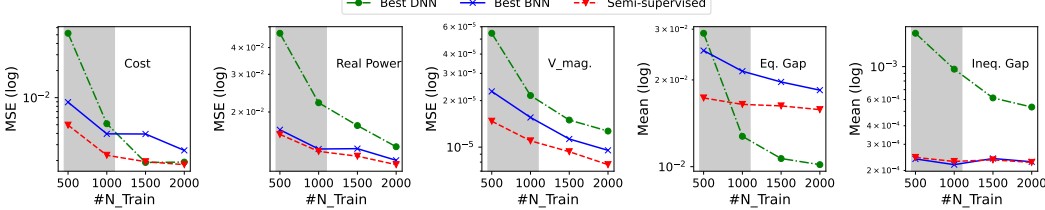

Figure 2: Comparative performance of DNN, BNN, and the proposed semi-supervised learning method across various training set sizes, evaluated by mean square error (MSE) and mean gap. The gray strip highlights the key training data range of 500 to 1000 samples. The semi-supervised method utilizing 20,000 unlabeled samples in $\mathcal{D}^f$, with a batch size of 1000 and $T_{max} = 1000$ seconds.

Before presenting further comparisons, we discuss the significance of the numerical errors and the potential improvements in the ACOPF problem. The cost values in ACOPF problems are in USD and mean value of cost for 57-Bus test case is $\$\,3.7 \times 10^4$ or 3.7 in per-unit system. Therefore, a mean error of 0.02 in per-unit system will imply the different \$200 across the testing instances. Further, in per-unit the voltage magnitude error requirement is below $10^{-5}$ as it will be equivalent of error 1 Volt for a 100 kilo-Volt system. More importantly, our target is to reduce the error values lower than the least count of the measuring instrument placed in the system to measure these quantities. Moreover, a 0.01 mean equality gap means that on average, 1.0 Megawatt of power imbalance occurs at a node.

We compare the proposed method's performance with various supervised and semi-supervised methods from [3] in Table 1, considering the target error discussion. It is clear that proposed method of Semi-supervised learning outperforms DNN method in terms of optimality and feasibility. Further, the objective gap and feasibility gaps are comparable using proposed approach even without the correction step involved in other state-of-art methods, (from [3] and [21]) in Table 1. Implication of the absence of correction step can be seen in the testing time[3], where the proposed approach and BNN have testing times similar to that of DNN while methods with correction step have one order of magnitude higher testing time. The reduction in testing time is crucial in the context of total time constrained situations which is the target application category for our BNN and Semi-supervised BNN based optimization proxies. The total time refers to the sum of the time required to obtain labeled dataset, training time and prediction time and is strictly limited in the case of ACOPF. The label generation time is reduced by using fewer supervised training samples and for the ACOPF, we constrain the training time to be $T_{max} = 1000$ sec [4]. The testing or prediction time will also be required to be as low as possible because we want to predict the solution of the ACOPF problem for a very large number of input instances in a given short time. This is crucial because one of the major application of these optimization proxies is in computing probabilistic estimates and the number of instances we can predict in a given time, will directly affect the accuracy of these estimates.

## 4   Conclusion and Future Works

The proposed Semi-supervised BNN has shown promise in working with low labeled dataset for constrained optimization problems. A major limitation is the higher time requirement to perform Bayesian inference, limiting the size of unlabeled dataset which can be used. Future work will involve scaling of the proposed scheme to larger size optimization problems, improving optimality-feasibility learning connections between *Sup* and *UnSup* blocks and exploiting BNN's predictive variance information for active learning.

**Broader Impacts:**

Improved solution of optimization problems will lead to more efficient resource utilization, benefiting industries by reducing costs and minimizing environmental impact. Further, improving ACOPF

---

[3]Time required to predict one single output given one testing input after model is trained i.e. time required for one forward pass

[4]Note that we are using unoptimized code without any GPUs which leaves potential to reduce further with optimized code and use of GPUs.

Table 1: Results on ACOPF over 100 test instances for 57-Bus. We compare the performance of the proposed method **without any projection** with 1000 labeled samples, with various existing methods from [3]. The optimality gap is from the optimizer solution with 0.949 sec. per sample solving time.

| Method | Correction | Obj. Gap | Mean Eq. | Mean Ineq. | Testing Time (s) |
|---|---|---|---|---|---|
| Proposed | No | 0.02 (0.00) | 0.01 (0.00) | 0.00(0.00) | 0.003 (0.000) |
| BNN | No | 0.04 (0.00) | 0.02 (0.00) | 0.00 (0.00) | 0.003 (0.000) |
| DC3 [3] | Yes | 0.01 (0.00) | 0.00 (0.00) | 0.00 (0.00) | 0.089 (0.000) |
| DC3, no soft loss [3] | Yes | 0.70 (0.05) | 0.07 (0.00) | 0.03 (0.01) | 0.088 (0.000) |
| Eq. NN [21] | Yes | 0.00 (0.00) | 0.00 (0.00) | 0.00 (0.00) | 0.039 (0.000) |

solution pipeline will directly help in combating climate change by optimizing the use of renewable energy and ensuring secure power grid operations [15].

**Acknowledgment:**

The authors acknowledge funding provided by LANL's Directed Research and Development (LDRD) project entitled "Science fAIr Project". The research conducted at Los Alamos National Laboratory is done under the auspices of the National Nuclear Security Administration of the U.S. Department of Energy under Contract No. 89233218CNA000001.

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

# Appendix

# A  Implementation Details

Table 2: Hyper-parameters and Implementation Details

| Hyper-parameter | DNN | BNN | Semi-supervised BNN |
|---|---|---|---|
| Learning Rate | $10^{-2}, \mathbf{10^{-3}}, 10^{-4}$ | $10^{-2}, \mathbf{10^{-3}}, 10^{-4}$ | $\mathbf{10^{-3}}$ |
| Decay Rate | $10^{-3}, \mathbf{10^{-4}}, 10^{-5}$ | $10^{-3}, \mathbf{10^{-4}}, 10^{-5}$ | $\mathbf{10^{-4}}$ |
| Batch Size (Sup) | 100 | 100 | 100 |
| Batch Size (UnSup) | – | – | 1000 |
| $T_{max}$ (sec.) | 1000 | 1000 | 1000 |
| $\sigma$ for $p(w)$ | – | $10^{-2}$ | $10^{-2}$ |
| Optimizer | Adam | Adam | Adam |
| Loss Function | MSE | TraceMeanELBO | TraceMeanELBO |

## B Additional Results

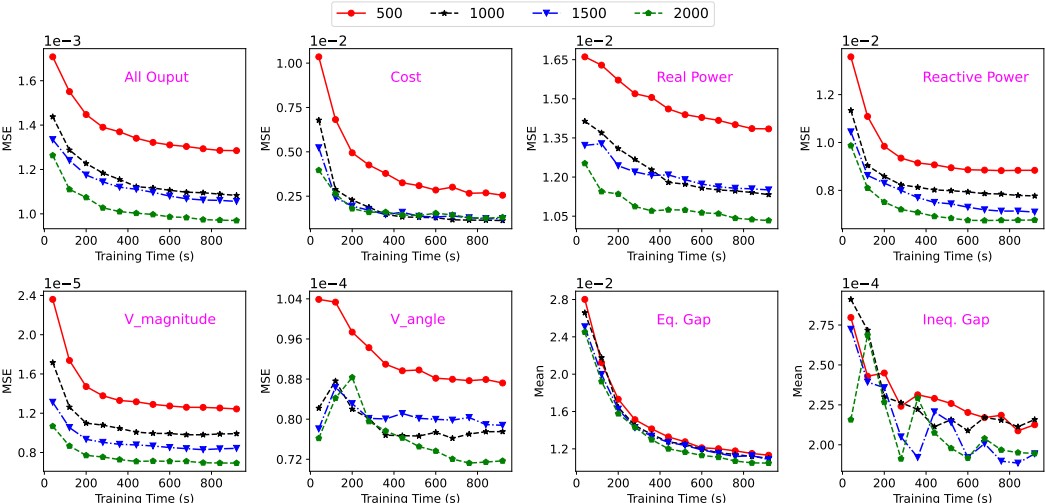

Figure 3: Performance of the Semi-supervised BNN on the 57-Bus ACOPF problem over training time. This figure illustrates the Mean Squared Error (MSE) and Mean Gap metrics for various outputs— aggregated output vector, objective value as cost, real power set-points, reactive power set-points, voltage magnitude, and voltage angle—plotted against the training time. The results offer insights into the effectiveness and efficiency of the semi-supervised BNN framework in solving the 57-Bus ACOPF problem.