# OpenReview forum: "Learning from Less: Bayesian Neural Networks for Optimization Proxy using Limited Labeled Data"
_NeurIPS.cc/2024/Workshop/BDU — NeurIPS BDU Workshop 2024 Poster_

### Official Review · Reviewer_mcpD · 2024-09-15
**Interesting idea and compelling experiments**

**Rating:** 7
**Confidence:** 4

**Review:**

**Summary:** The authors propose a semi-supervised training paradigm to learn ML proxies for optimization problems in settings under budget constraints corresponding to training time along with minimal labelled data. Specifically, the authors aim to replace constrained optimization algorithms using Bayesian Neural Networks.

---

**Strengths:**
- The authors propose a feasibility-based data augmentation scheme to aid learning using unlabelled data for developing optimization proxies. It is interesting and novel.
- The authors also present compelling results demonstrating the proposed method's efficacy.
- They also identified and addressed the limitations of the proposed method.

---

**Weakness**
- Presentation is weak and could be improved, but it may be due to the choice of the 4-page format.

---

The work presented is compelling and deserves an opportunity to be showcased in the workshop.

---

### Official Review · Reviewer_cQFd · 2024-09-24
**Review of "Learning from Less: Bayesian Neural Networks for Optimization Proxy using Limited Labeled Data"**

**Rating:** 7
**Confidence:** 1

**Review:**

I will first note that I have no background in the field of BNNs, so my understanding of this work was limited.

Pros:
- Proposes a semi-supervised BNN for solving optimization problems with equality and inequality constraints.
- Includes a short discussion of other approaches in this area and why they believe this approach is a better fit for this specific problem.
- Clearly lays out the problem setup and describes the dataset used for numerical experiments.
- Plots are clear and illustrate how the proposed method does well in the regime of interest (limited labeled samples)
- Exposition is concise and gets the point across.
- Highlights other metrics of interest like testing time.
- Compares against both standard BNNs and DNNs.

Cons:
- Several parts of the paper were unclear to me:
	- The description of the unsupervised step was not completely clear. For example, when constructing the feasibility dataset they say
	  > Under the consideration that for each input there exist a solution of (1), we can argue that for each input the feasibility gap function (2) has optimal value or true label of 0. We can augment the unlabeled dataset $D^u$ such that it becomes a labeled feasibility dataset i.e. $D^f = \{ x_j , 0 \}^M_{j=1}$.

	  This doesn't really make sense to me. The feasibility gap function is a function of both an input $x$ and a solution $y$, not only of the input $x$. So how does it make sense to label each $x$ with a label of $0$ for the feasibility loss?
	- The paragraph starting with the sentence "Before presenting further comparisons, we discuss the significance of the numerical errors and the 111 potential improvements in the ACOPF problem." felt somewhat out of place, and it was not obvious to me what the takeaway was for me in regards to what followed (the comparison of the semi-supervised BNNs performance for this problem).

---

### Decision · Program_Chairs · 2024-10-09

Accept (Poster)